# Effects of Extraction and Evaporation Methods on Physico-Chemical, Functional, and Nutritional Properties of Syrups from Barhi Dates (*Phoenix dactylifera* L.)

**DOI:** 10.3390/foods12061268

**Published:** 2023-03-16

**Authors:** Kanokporn Julai, Pimnapanut Sridonpai, Chitraporn Ngampeerapong, Karaked Tongdonpo, Uthaiwan Suttisansanee, Wantanee Kriengsinyos, Nattira On-Nom, Nattapol Tangsuphoom

**Affiliations:** 1Master of Science Program in Food Science for Nutrition (International Program), Institute of Nutrition, Mahidol University, Salaya, Phutthamonthon, Nakhon Pathom 73170, Thailand; kanokporn.jul@student.mahidol.ac.th; 2Food and Nutrition Academic and Research Cluster, Institute of Nutrition, Mahidol University, Salaya, Phutthamonthon, Nakhon Pathom 73170, Thailand; pimnapanut.sri@mahidol.ac.th (P.S.); karaked.ton@mahidol.ac.th (K.T.); uthaiwan.sut@mahidol.ac.th (U.S.); wantanee.krieng@mahidol.ac.th (W.K.); nattira.onn@mahidol.ac.th (N.O.-N.); 3Food Science and Technology Division, Faculty of Engineering and Agro-Industry, Maejo University, Nong Han, San Sai, Chiang Mai 50290, Thailand; chitraporn_n@mju.ac.th

**Keywords:** date fruits, syrup, pectinase, cellulase, glycemic index

## Abstract

Date fruits (*Phoenix dactylifera* L.) are rich in sugar and also contain a substantial amount of phenolic compounds. Therefore, date fruits can be used to produce an alternative sweetener, having lower glycemic index than sucrose. This study investigated the effects of extraction and evaporation methods on various properties of the syrups prepared from Barhi dates. Extraction of date juice with the aid of pectinase or cellulase significantly enhanced the production yield, total phenolic content, and antioxidant activities determined by Ferric Reducing Antioxidant Power and Oxygen Radical Absorbance Capacity assays. Syrups prepared without enzyme application had about 6–7 times higher apparent viscosity than those prepared from the enzyme-assisted extracted juices. Vacuum evaporation produced syrups with significantly lighter color and inferior antioxidant properties than open heating. Properties of date syrups prepared with or without enzyme-assisted extraction followed by open heat evaporation were not different. They had a glucose-to-fructose ratio close to 1:1, received good sensory acceptability scores of above 6 on a 9-point hedonic scale, contained a safe level (<40 mg/kg) of 5-hydroxymethyl-2-furfuraldehyde, and exhibited similar glass transition and melting temperatures; while a greater inhibition on α-amylase activity was observed in syrups obtained from enzyme-assisted extraction. The in vivo glycemic measurement revealed that the syrup prepared with the aid of Pectinex and open heating was classified as low glycemic index (GI = 55) and medium glycemic load (GL = 11). Thus, enzyme-assisted extraction of date juice using Pectinex could be used to produce a healthy natural sweetener from low quality date fruits.

## 1. Introduction

The increasing prevalence of non-communicable diseases, particularly diabetes and obesity, has become a major global public health concern. According to the World Health Statistics report in 2022, the global share of deaths attributable to non-communicable diseases was 74%, and the global prevalence of obesity was markedly higher among adults aged 15–49 years [1]. It is evident that the excessive consumption of sugar is linked with the higher risk of obesity and type 2 diabetes. The largest source of added sugar comes from processed food, especially from sweetened non-alcoholic beverages. A recent survey suggested that the average daily sugar intake of Thai people was about four times higher than the daily recommendation of six teaspoons. Regarding this, the demand for low glycemic sweeteners is increasing; hence, there are currently an abundance of reduced- and low sugar food products in the markets. However, many of these contain synthetic or artificial sweeteners that may be of concern for some consumers regarding the possible long-term side effects [2]. Therefore, the development of alternative natural low-glycemic sweeteners, which can better replace the regular table sugar in food products, has gained more interest from the food industry. In regard to this, sweeteners derived from the palm family (Arecaceae) been studied and recognized for their greater bioactive properties and lower glycemic index than table sugar [3,4,5,6].

Date palm (*Phoenix dactylifera* L.) is one of the important subsistent crops in many countries. The plant originated from warm desert areas of the Middle East region and North Africa [7]. Nowadays, date fruit is marketed globally as high-value fruit due to the role of functional and medicinal food crop. The world’s production amount of date fruit reached 9.45 million metric tons in 2020 [8]. The growing areas of date palm have been expanded during the past decade to several tropical Asian countries. In Thailand, the most widely grown date palm is Barhi variety, of which the fruits are consumed in fresh form at Khalal and Rutab maturation stages, i.e., 120 and 135 days after pollination, respectively. Date fruits are a source of many nutrients and functional compounds. Date flesh contains a high amount of sugar, especially fructose, glucose, and sucrose, as well as dietary fiber and phenolic compounds [7,9]. Nowadays, there are significant quantities of date fruits that failed to meet the specification or standard for selling. Those low-quality date fruits are sold at a low price or even discarded, even though they are safe for human consumption, and have similar composition to the normal ones [10]. The transformation of low-quality dates into other food products, such as nectars, juices, jellies, jams, and frozen pulp, has been previously reported elsewhere [11].

Owing to their composition, dates have been used for the preparation of syrup and are considered as an alternative natural liquid sugar. Typically, syrup production from fruits consists of extraction and concentration of fruit juice or sap. An efficient extraction process is very important to the yield and quality of the obtained juice. Enzyme-assisted extraction is one of the alternative methods to improve the efficiency of fruit juice extraction, compared with the extraction using only conventional heating or mechanical pressing [12]. Pectinases convert insoluble pectic substances into more soluble pectin; thus, improve the efficiency of extraction and the clarity of fruit juice. Cellulase breaks down insoluble cellulose into shorter chain substances with increasing solubility, which helps soften the plant tissues and increases the juice extraction yield [13]. In addition, pectinase and cellulase enzymes have been found to improve phenolic content recovery and antioxidant properties of fruit juices by enhancing the release of the trapped bioactive compounds [14]. The concentration step generally uses the evaporation technique to reduce the moisture content of the extracted juice to comply with the standard specifications [15]. However, many studies reported that the heat applied during conventional evaporation by open heating can cause a change in the nutrients and sensory quality of the obtained syrup [16]. Upon heating, several substances are formed, especially by the Maillard reaction and caramelization of sugar, resulting in the darkened color, as well as the increasing antimicrobial, antiallergenic, and antioxidant activities. Among those substances, 5-hydroxymethyl-2-furfuraldehyde (HMF) may form by the dehydration of reducing sugars. Nevertheless, high intake of HMF (30–150 mg/d) can cause adverse effects on human health, i.e., mutagenic, genotoxic, cytotoxic, and enzyme inhibitory. Evaporation at lower temperatures under a vacuum has been demonstrated to improve the quality and better preserve the quality and antioxidant properties in syrups prepared from saps and fruit juices [16].

Though the processing and properties of date syrups have been reported elsewhere, most studies used date fruits of other varieties that are consumed as dry, fully mature fruits at the Tamar stage (150 days post-pollination) [17,18,19,20]. Our previous study found that date palm fruits at the Khalal stage of Barhi cultivar were rich in antioxidants and possessed potent inhibitory effects on key enzymes governing various non-communicable diseases [21]. Therefore, this study aimed to investigate the effects of juice extraction and evaporation methods on the physico-chemical, functional, nutritional, and sensory properties of the syrup prepared from Barhi dates. An in vivo test in human subjects was also conducted to determine the glycemic properties of the obtained syrup. The information would be useful for the selection of processing methods in the production of an alternative natural sweetener from dates with improved quality.

## 2. Materials and Methods

### 2.1. Materials and Reagents

Fruits of date palm (*Phoenix dactylifera* L.) of Barhi cultivar were donated by Aura International Co., Ltd. (Kanchanaburi, Thailand). The fruits were harvested at Khalal maturity stage (120–135 days pollination) from culture-originated female palm trees grown in Kanchanaburi province, Thailand. After harvesting, the fruits were sorted according to the company’s quality specification, which is consistent with the international standard for dates [22]. Low quality date fruits, including those falling off the bunch, having brown spots on skin, too small size, too soft texture, and being overripe, were collected and kept refrigerated at 4 °C for less than 72 h before being transported under ambient conditions to the laboratory. After receiving, the fruits were kept refrigerated at 4 °C for less than 24 h, before being parted from bunches, washed under running tap water, and the surface wiped dry. The cleaned date fruits were vacuum-packed in low-density polyethylene plastic bags and stored frozen at −20 °C until being used. Based on the chemical composition analysis, which was determined according to the official methods of AOAC 2019 [23] from 3 harvesting batches, 100 g of the fresh date flesh consisted of 68.98 ± 0.69 g moisture, 0.83 ± 0.02 g protein, 0.12 ± 0.10 g fat, 1.07 ± 0.07 g ash, 29.00 ± 0.74 g total carbohydrate, 5.15 ± 0.05 g dietary fiber, and 23.02 ± 0.30 g total sugar.

Commercial hydrolytic enzymes, namely pectinase mixture, Pectinex^®^ Ultra SP-L (3300 PGNU/g; Novozymes A/S, Bagsværd, Denmark), and cellulase mixture, Viscozyme^®^ L (100 FBG/g; Novozymes A/S), were donated by Brenntag Ingredients (Bangkok, Thailand). All chemicals and reagents, unless stated otherwise, were obtained from Sigma-Aldrich (St. Louis, MO, USA).

### 2.2. Preparation of Date Syrup

#### 2.2.1. Juice Extraction

The frozen date fruits were thawed, halved, and depitted manually, prior to homogenizing with deionized water at the weight-to-volume ratio of 1:2 using an electric blender (series MMB12P4RGB, Bosch, Bangkok, Thailand) to obtain date pulp. Enzyme-assisted extraction of date juice was performed according to the suitable conditions obtained from our preliminary experiments. The date pulp was treated either with Pectinex^®^ Ultra SP-L or Viscozyme^®^ L at the enzyme-to-substrate ratio of 50 U/100 g date flesh. The mixture was mixed thoroughly with a magnetic stirrer and placed in a thermostatically controlled shaking water bath at 50 °C, which is the enzyme’s optimum temperature, for 2 h for Pectinex, or 4 h for Viscozyme. Upon completion, the pulp was heated at 90 °C for 5 min in a boiling water bath to inactivate the enzyme, prior to pressing with a press juicer at 1 MPa pressure, and filtering through a double layer cheesecloth to obtain date juice. Conventional extraction of date juice was performed without the application of enzymes by heating the date pulp at 95 °C for 15 min, followed by pressing and filtering, as described earlier. The extraction was conducted in triplicate and the soluble solids yield of each of the obtained juices was calculated from its total soluble solids content (TSS), which was measured using a hand refractometer (0–32° Brix, ATAGO, Tokyo, Japan).

(1)
$$Soluble\;solids\;yield\;(\%)\;=\;\frac{Weight\;of\;date\;juice\;\times \;TSS\;of\;date\;juice}{Weight\;of\;date\;flesh}\;\times \;100$$


#### 2.2.2. Evaporation

Evaporation of date juice was performed by open heating or vacuum evaporation, as described elsewhere [24], to obtain syrups with the total soluble solids of 72° Brix, to comply with the Thai standard for plant syrups. Open-heat evaporation was performed by heating the date juice at 95 °C for 3 h on a hotplate (C-MAG HS7, IKA, Staufen, Germany) with constant agitation using an overhead stirrer (MS-40, MIULAB, Zhejiang, China). Vacuum evaporation was achieved using a rotary evaporator (R-114RE B-480, Buchi, Flawil, Switzerland) at 50 °C, with respective vacuum pressure of 16 kPa for 25 min. Total soluble solids content of the syrups was monitored using a hand refractometer (58–90° Brix, ATAGO, Tokyo, Japan). The obtained syrups were hot filled in glass jars, cooled down and stored at room temperature for analyses. Syrup samples prepared from date juice extracted without the enzyme and concentrated by open heating and vacuum evaporation are referred to as CO and CV, respectively. The samples involving Pectinex-assisted extraction are labelled as PO and PV, while those with Viscozyme are mentioned as VO and VV. All syrup samples were prepared in triplicate for further analyses. The production yield of syrup was calculated as the percentage of the amount of date syrup obtained from 100 g date flesh, as follows:
(2)
$$Production\;yield\;\left(\%\right)=\frac{Weight\;of\;date\;syrup}{Weight\;of\;date\;flesh}\;\times \;100$$


### 2.3. Analyses

#### 2.3.1. pH Value

pH value was measured at 25 °C based on AOAC method 943.02 [23] using a digital pH meter (Denver Instrument, Denver, CO, USA) with a glass electrode. Measurements were conducted in triplicate.

#### 2.3.2. Water Activity

Water activity (a_w_) was measured based on AOAC method 978.78 [23] using a portable water activity analyzer (Novasina ms1, Axair, Pfäffikon, Switzerland) at a controlled temperature (25 ± 1 °C). Measurements were conducted in triplicate.

#### 2.3.3. Color Values

The color of samples was measured as Hunter Lab values (L*, a*, b*) by using a spectrocolorimeter (ColorFlex EZ, Hunter Associates Laboratory, Reston, VA, USA). L* value indicates lightness or brightness; a* value indicates red for positive value and green for negative value; b* value indicates yellow for positive value and blue for negative value. Measurements were conducted in triplicate. The results were also expressed as browning index (BI) to indicate the extent of browning, which were calculated as follows [25]:
(3)
$$BI\;=\;100\;\times \;\frac{X\;-\;0.31}{0.17}$$

where:
(4)
$$X\;=\;\frac{a*\;+\;1.75\;L*}{5.645\;L*\;+\;a*\;-\;3.012\;b*}$$


In addition, the total color difference (∆E*) were calculated to determine the difference in color of syrup samples, using the following equation [26]:
(5)
$$\Delta E*\;=\;\sqrt{{\left(L_{2}^{*}\;-\;L_{1}^{*}\right)}^{2}\;+\;{\left(a_{2}^{*}\;-\;a_{1}^{*}\right)}^{2}\;+\;{\left(b_{2}^{*}\;-\;b_{1}^{*}\right)}^{2}}$$

where L*_1_, a*_1_ and b*_1_ were the values of date syrup prepared without enzyme followed by evaporation using open heating (CO), and L*_2_, a*_2_ and b*_2_ were the color values of other date syrups.

#### 2.3.4. Rheological Properties

Rheological measurements of syrup samples at room temperature (25 ± 0.1 °C) were carried out according to the previously described methods with slight modifications [27] using a rheometer (HAAKE MARS 40, Thermo Fisher Scientific, Karlsruhe, Germany) fitted with a cone and plate geometry probe (C35 2°/Ti, gap 0.100 mm) and a peltier temperature control. A steady shear rate sweep was performed by increasing shear rate from 0.1 to 100 s^−1^ at constant temperature. Measurements were conducted in triplicate. Apparent viscosity (mPa·s) at the shear rate of 50 s^−1^ was reported. Flow behavior and consistency indices were determined by fitting the obtained flow curves using the power law model [28]:
(6)
$$\tau \;=\;K\dot{\gamma }n$$

where τ is shear stress (Pa), 
$\dot{\gamma }$
 is shear rate (s^−1^), K is the consistency index (Pa · s^n^), and n is the flow behavior index.

#### 2.3.5. Antioxidant Properties

Total phenolic content (TPC) and antioxidant activities of date syrup samples were analyzed using well-established methods, as described in a previous work [21]. The date syrup was diluted in deionized water at an appropriate ratio, filtered through a 0.45 µM PES membrane syringe filter, and kept frozen at −20 °C until analyzed. Total phenolic content (TPC) was determined using Folin–Ciocalteu’s phenol reagent and the absorbance was measured at 765 nm using a microplate reader (Synergy HT, Bio-Tek Instruments, Winooski, VT, USA) at 25 °C. Gallic acid solution (0–200 µg/mL) was used as a standard. TPC value was calculated as gallic acid equivalent (mg GAE/100 mL). Antioxidant activities were determined using 2,2-diphenyl-1-picrylhydrazyl (DPPH) radical scavenging, ferric ion reducing antioxidant power (FRAP), and oxygen radical absorbance capacity (ORAC) assays. For the DPPH assay, the absorbance was measured at 520 nm using a microplate reader at 25 °C. Trolox solution (0–0.64 µM) was used as a standard and DPPH value was expressed as Trolox equivalent (µmol TE/100 mL). For the FRAP assay, absorbance was measured at 595 nm using a microplate reader at 37 °C. Trolox solution (0–250 µM) was used as a standard. The FRAP value was expressed as the Trolox equivalent (µmol TE/100 mL). For the ORAC assay, the fluorescence was recorded using a microplate reader at 37 °C with constant shaking at 1 min intervals for 90 min at excitation and emission wavelengths of 485 and 528 nm, respectively. The ORAC value was calculated based on the differences in areas under the sodium fluorescein decay curve and expressed as µmol TE/100 mL. Analyses were conducted in triplicate.

#### 2.3.6. Chemical Composition

Nutritional composition of date syrups was analyzed according to the AOAC official methods [23]. Moisture content was determined using the gravimetric method (AOAC 945.62). Protein content was analyzed by the Kjeldahl method (AOAC 945.23), in which the nitrogen content was multiplied by the conversion factor of 6.25. Total fat was analyzed by acid hydrolysis of the sample followed by extracting the dried sample with petroleum ether in the Soxhlet apparatus (AOAC 2000.18). Total ash was analyzed by incineration in a muffle furnace at 450 °C (AOAC 945.28). Total carbohydrate was calculated by subtracting the percentage of moisture, crude protein, crude fat, and ash from 100. Dietary fiber was analyzed by enzymatically gravimetric method, including soluble and insoluble dietary fibers (AOAC 991.42). Sugar content, including glucose, fructose, sucrose, and maltose was determined using an HPLC system equipped with Alltech^®^ 800 ELSD detector (Buchi, New Castle, DE, USA) and a prevailing carbohydrate column (5 µm, 250 mm × 4.6 mm internal diameter; Asahipak NH2P-50 4E, Shodex, Kanagawa, Japan) (AOAC 982.14). Energy content was calculated by multiplying the content of each macronutrient with its caloric content, i.e., 4 kcal/g for protein and carbohydrate, and 9 kcal/g for fat. Analyses was carried out in duplicate of each syrup sample.

Analysis of 5-hydroxymethyl-2-furfuraldehyde (HMF) in date syrups was carried out based on the method of AOAC [23]. Syrup samples were diluted in deionized water at an appropriate ratio and filtered through a 0.45 μm filter, prior to injecting immediately into an HPLC equipped with a diode array detector, and a Shim-pack GIST C18 column (5 μm, 150 mm × 4.6 mm internal diameter; Shimadzu, Kyoto, Japan), fitted with a guard cartridge. HMF was identified by splitting the peak with a standard HMF and by comparing the HMF spectrum with that of the syrups. Analyses were performed in duplicate.

#### 2.3.7. Inhibitory Activities on Carbohydrate-Digestive Enzymes

The inhibitory activities of α-amylase and α-glucosidase were investigated using the protocols previously described [21]. The α-amylase activity assay used porcine pancreatic α-amylase (type VII, ≥10 unit/mg) as an enzyme, while the α-glucosidase activity used *Saccharomyces cerevisiae* α-glucosidase (type 1, ≥10 U/mg protein). The substrate in both assays was *p*-nitrophenyl-*α*-*D*-maltopentaoside. The enzyme inhibitory activities were determined by measuring the absorbance at 405 nm using a 96-well UV-visible microplate reader, and calculated as a percentage of inhibition:
(7)
$$Inhibitory\;activity\;\left(\%\;inhibition\right)\;=\;\left(1\;-\;\frac{B\;-\;b}{A\;-\;a}\right)\;\times \;100$$

where A is an initial velocity of the control reaction with enzyme (control), a is an initial velocity of the control reaction without enzyme (control blank), B is an initial velocity of the enzyme reaction with extract (sample), and b is an initial velocity of the reaction with extract but without enzyme (sample blank). The inhibition was reported as % inhibition of 5 µL syrup in 200 µL assay. Analyses were performed in triplicate.

#### 2.3.8. Thermal Properties

Thermal measurements of syrup samples were carried out using a differential scanning calorimeter (DSC 214 Polyma, NETZSCH, Bayern, Germany). The calorimeter was calibrated with indium standards. Samples (3–4 mg) were weighed into aluminum pans, and hermetically sealed. A hermetically-sealed empty aluminum pan was used as a reference. In order to avoid condensation of water, nitrogen gas was used to purge the furnace chamber at 80 mL/min outside and 60 mL/min inside. The thermal scans were conducted from –65 to 160 °C at a heating rate of 10 °C/min [29]. At least three DSC runs were carried out for each sample. Glass transition temperature (*T_g_*), which is the temperature, at which inflection point in the specific heat capacity occurred in order to form an amorphous glass, as well as melting temperature (*T_m_*), were determined using a thermal analysis software (Proteus^®^ version 8, NETZSCH, Bayern Germany). The temperatures at peak values are reported. 

### 2.4. Sensory Evaluation

Sensory evaluation was performed by 60 untrained panelists aged 18 years old and above, who were recruited from staff and students of Mahidol University to represent the general population who are target consumer of the syrup product. The protocol was approved by the Mahidol University Central Institutional Review Board. Date syrup samples (10 g) were put in clear plastic cups covered with lids and labelled with 3-digit random numbers. Panelists were asked to taste the samples at room temperature together with plain pancakes in a monadic sequential presentation scheme, randomly assigned for each panelist. Each sample was evaluated for appearance, color, odor, consistency, taste, and overall acceptability using a 9-point hedonic scale (9 = like extremely, 5 = neither like nor dislike and 1 = dislike extremely). Panelists were asked to rinse their mouths with bottled drinking water between date syrup samples [30].

### 2.5. In Vivo Study

The glycemic index (GI) of date syrup PO was evaluated following the ISO 26642 method [31]. Ethical approval was obtained from the Mahidol University Central Institutional Review Board. Twelve male and female subjects were recruited to participate in the study. The inclusion criteria for participation were that subjects should be healthy males or females, aged between 20 and 45 years, with a body mass index (BMI) ranging from 18.5 to 22.9 kg/m^2^. Exclusion criteria were abnormal fasting blood sugar ≥100 mg/dL, known diabetes mellitus and related conditions, major medical or surgical events requiring hospitalization within the past 3 mo, being pregnant, breastfeeding, and use of medications affecting glucose tolerance and gastrointestinal functions. Each subject gave written informed consent for the study and were verbally briefed about the assessment. The weights and heights of subjects were obtained following standard procedures.

Date syrup was prepared under hygienic conditions within 1 wk prior to the experiment, packed in capped glass jars, and stored at room temperature. The syrup was tested for aerobic plate count, yeasts and molds, *Salmonella* spp., and *Staphylococcus aureus* to ensure the compliance with microbial standard. The available carbohydrate content in the tested sample was determined by subtracting the carbohydrate content with the dietary fiber content, which were analyzed according to the aforementioned AOAC official methods [23].

Each subject completed the set of four weekly visits, including three visits to obtain the incremental area under the curve (IAUC) for the reference food (anhydrous glucose) and one for the testing date syrup in a random sequence. The day before the study, each participant was informed refrain from vigorous exercise and consumption of beverages containing alcohol or caffeine. The subjects were asked to consume a standard dinner and observe a 10–12 h overnight fast preceding the test. On the test day, an intravenous catheter was inserted into an antecubital vein for blood sampling and a fasting blood sample was drawn. Subjects were required to consume 200 mL of reference food or test food solution containing 50 g of available carbohydrate, i.e., 50 g of anhydrous glucose or 70.16 g of date syrup, which was prepared by dissolving the food in drinking water. Each subject was asked to finish the solution within 10 min, followed by 100 mL of drinking water. Subjects are required to remain sedentary during 3 h of the experiment, and were prohibited from eating and drinking until the end of the session.

Blood samples were taken at 15, 30, 45, 60, 90, and 120 min after complete ingestion of the meal. Blood samples were centrifuged to separate plasma, and the plasma glucose (mg/dL) was analyzed by the glucose oxidase method using a commercial kit with an automatic biochemistry analyzer (BioSystems, Barcelona, Spain). The IAUC for reference food and date syrup was calculated using GraphPad Prism 5.01 software (GraphPad Software Inc., San Diego, CA, USA). The GI value of date syrup was calculated from IAUC of the test food and the average IAUC (n = 3) for the reference food in each subject, as follows:
(8)
$$GI=\frac{Blood\;glucose\;IAUC\;for\;test\;food}{Average\;IAUC\;for\;the\;reference\;glucose}\;\times \;100$$


The glycemic load (GL) was calculated from the amount of available carbohydrates in a standard serving of syrup, i.e., 30 g or 2 tablespoons, and the GI.

(9)
$$GL\;=\;\frac{GI\;\times \;Available\;CHO\;in\;a\;serving\;size\;(g)}{100}$$


### 2.6. Statistical Analysis

The data was analyzed by IBM SPSS Statistics for Windows Version 18.0 (IBM, Armonk, NY, USA). The data was presented as means and standard deviations, except that chemical composition was reported as means of duplicate analysis. Mean difference was carried out by independent sample t-test or one-way Analysis of Variance (ANOVA) and the differences between means were distinguished using Duncan’s Multiple Range Test at *p* < 0.05.

## 3. Results and Discussion

### 3.1. Production Yield

The date juice extracted without the addition of enzymes had the lowest soluble solids yield (54.27 ± 5.89%). The soluble solids yield obtained from Pectinex- and Viscozyme-assisted extractions were 71.39 ± 0.66% and 73.29 ± 1.22%, respectively, which were significantly higher than that extracted in the absence of enzymes. There was no significant difference between the soluble solids yield of extraction using Pectinex and Viscozyme. Enzyme-assisted extraction increased the juice extraction efficiency by hydrolyzing the structural polysaccharides in plant tissues, resulting in the tissue softening. Such phenomenon led to the greater release of intracellular components, particularly soluble substances, such as sugars and organic acids, hence the soluble solids yield of the obtained juice increased. Several researchers have reported that pectinolytic and cellulolytic enzymes enhanced the liberation of sugar and phytochemical compounds during the extraction, resulting the improved physico-chemical and functional properties of the extracted juice [17,32].

The production yield of date syrup depended largely on the juice extraction method (Figure 1). Enzyme-assisted juice extraction increased the production yield of date syrup from below 30% to about 40%, compared with those prepared without enzyme. Among the syrups prepared using different enzymes, the production yield was similar. Different evaporation methods also did not significantly affect the syrup production yield.

### 3.2. Physicochemical Properties

All date syrups had pH values ranged from 4.80–5.39, which were categorized as low acid food. Syrups prepared from any enzyme-extracted juices had pH values of lower than 5, which were lower than those prepared without enzyme. The pH of date syrups subjected to different evaporation methods was similar. The more acidic syrups produced from enzyme-extracted juices was due to the significantly increased amount of organic acids being liberated from the fruit tissue during enzyme-assisted extraction [32]. The a_w_ of all syrup samples prepared using different juice extraction and evaporation methods was about 0.60, owing to their similar TSS. It is known that food products with intermediate a_w_ of 0.6–0.7 are highly susceptible to Maillard browning reaction [33], especially upon heating.

Color is one of the quality indices of syrups, which is often associated with the chemical composition and content, as well as their changes upon the processing. Photographs taken of date syrups prepared using different extraction and evaporation methods are presented in Figure 2. Color values of date syrups depended largely on the conditions used for juice evaporation, and to a smaller extent on the juice extraction method. Syrups evaporated using open heating had lower L* and b* but higher a* values than those evaporated under vacuum (Table 1). The L* value of the syrups evaporated under vacuum was about two times higher than those evaporated using open heating. When the syrups prepared using the same evaporation condition were compared, it was observed that syrups from both enzyme-extracted juices had higher L* but lower a* and b* than those prepared from the juice extracted without enzyme. Among the syrups prepared from enzyme-assisted extraction, the color values of syrups obtain from similar evaporation condition were about similar, except that the a* values of syrup involving juice extraction using Pectinex were slightly higher than their counterparts. The findings suggested that higher temperature used for evaporation led to the darker color of syrup. In a previous study, L*, a*, and b* values approaching zero were reported for date syrups produced from Barhi and Safri varieties at the Tamar maturity stage [34]. It was likely that the syrup had a dark color because date fruits at the Tamar stage had brownish flesh. On the other hand, the flesh of Barhi date fruits at Khalal stage used in this study was yellow in color, resulting in the syrup with a lighter color.

The BI of syrups reflected the extent of the browning reaction, which occurred during juice extraction and evaporation. Syrups obtained from vacuum evaporation had lower BI than those evaporated by open heating. Such differences contributed to their different color values and were visually noticeable from the syrups’ brown color. It was plausible that the harsher condition (higher temperature and longer time) of open heating, as well as the exposure to oxygen, led to the greater extent of browning reactions [35]. Thus, the open-heated syrups had a more intense brown color than those evaporated under a vacuum due to the increased production of Maillard reaction products and caramelization products [36,37]. A previous study on coconut sap sugar also found a significant effect on the color of finished product when high temperature and long duration of evaporation process were used [24]. The lower pH values of syrups prepared with the aid of enzymes could also retard the browning reaction, resulting in the lower BI. The ΔE* values described the extent of the difference in color of date syrups against the syrup prepared using conventional extraction and evaporation methods (CO). Syrups prepared using enzyme-assisted extraction and open heating evaporation had ΔE* values between 3.5 and 5, indicating that clear difference in color was noticed [26]. This was consistent with their lighter color than CO. For all syrups evaporated under vacuum, the ΔE* values of exceeded 20, which revealed that they had different color to CO, i.e., a much lighter color.

Rheological properties affect the flow characteristics, particularly the pourability and spoonability, of honey and honey-like products [38]. The rheograms of date syrups prepared using different conditions for juice extraction and evaporation are presented as Figure 3. The rheological measurements were conducted at room temperature to resemble the temperature at which syrups and honey are normally consumed. All date syrups were non-Newtonian, pseudoplastic fluids (Table 2).

Viscosity is important in the production and the application of syrups in food. Highly viscous syrups would be difficult to pour and mix into the food, while syrups with too low viscosity would be too runny to cling on the top of the food surface. Apparent viscosity at the shear rate of 50 s^−1^ was compared because such shear rate can represent the oral processing. Overall, date syrups prepared without enzymes had up to 10 times higher viscosity than those prepared with enzyme application for date juice extraction. Pecinex-assisted extraction gave syrups a higher viscosity than Viscozyme, regardless of the evaporation conditions. The evaporation method less affected the viscosity of the obtained syrup than the extraction method. When the syrups evaporated under different conditions were compared, it was found that syrups extracted without enzyme and evaporated with open heating were more viscous than its counterpart evaporated under vacuum. For Pectinex and Viscozyme-extracted juice, the evaporation condition did not affect the viscosity of the obtained syrups. The lower apparent viscosity of the syrup produced with the aid of enzymes occurred from the degradation of the structural polysaccharides naturally present in the fruit cell wall and the reduced complex interactions among soluble sugars, pectin substances, and suspended solids [39]. Considering the liquid category of the United States’ National Dysphagia Diet [40], date syrups prepared without enzymes were spoon-thick liquid (>1750 mPa·s) at a shear rate 50 s^−1^. On the other hand, date syrups prepared with Pectinex and Viscozyme were honey-like (351–1750 mPa·s) and nectar-like (51–350 mPa·s), respectively.

Flow behavior index and consistency index were obtained by fitting the flow curves with the power law model. The coefficient of determination (R^2^) for the fitted curves of all samples was 0.9718–0.9997. The flow behavior index of date syrup samples was 0.40–0.88, indicating that all syrups were non-Newtonian fluid. In addition, their flow curves (Figure 3) demonstrated the shear thinning behavior, in which the viscosity decreased with the increasing shear rates [28]. Flow behavior of date syrup has been found to associate with its concentration, thus, syrups with lower total soluble solids tended to exhibit the more Newtonian behavior [41]. The consistency index of all samples was consistent with their viscosity that the more viscous syrups had the higher consistency index (Table 2). Enzyme-assisted extraction of date juice gave the syrups that were less viscous and exhibited lower flow behavior index than the juice extracted without enzymes. This could be explained by the fact that the syrups prepared without the aid of enzymes might contain a larger amount of long chain polysaccharides, which could form a network with other components in the syrups. In particular, cellulose is known to interact well with water through hydrogen bonding [42]. The greater presence of such network might lead to the shear thinning effect due to the deformation of the network induced by the applied shear force [43]. Another plausible explanation was that the brown pigment polymers from the browning reaction might increase the viscosity and shear thinning behavior of the syrups. The lower viscosity and flow behavior index of date syrups prepared by enzyme-assisted extraction of date juice might be advantageous for their application in food. Syrups with lower viscosity would have greater pourability and spreadability; hence flow more easily under force but at the same time could hold its shape when not subjected to any external force other than gravity [44]. Thus, the date syrups could be applied directly as sweetener in food and beverages, in which the syrup would dispersed quickly and readily.

### 3.3. Antioxidant Properties

Date syrups prepared from enzyme-extracted date juice had higher TPC than those extracted without enzymes when the same evaporation condition was used (Table 3). Considering the effect of the evaporation method, syrups evaporated by open heating contained higher TPC than those evaporated under a vacuum at lower temperatures. The DPPH values of date syrups prepared from enzyme-extracted juice and evaporated with open heating were higher than those evaporated under a vacuum. However, for syrups prepared in the absence of enzymes, the DPPH were similar for all evaporation conditions. The higher FRAP values were observed in syrups from enzyme-extracted juice than those extracted without enzymes, though the difference was to a lesser extent than TPC and DPPH. Evaporation using open heating also resulted in syrups with higher FRAP values than vacuum evaporation. For the ORAC value, a similar trend to TPC and FRAP values was observed. Enzyme-assisted extraction resulted in date syrups with higher ORAC than those prepared without enzymes. Moreover, evaporation using open heating also produces syrups with higher ORAC than vacuum evaporation. Pectinex-assisted syrup exhibited a higher ORAC value for similar evaporation conditions than its counterparts.

The application of enzymes in date juice extraction increased the TPC and antioxidant activities, measured by FRAP and ORAC assays, of the obtained date syrups. This was because Pectinex and Viscozyme facilitated the degradation of the cell middle lamella and primary wall, thereby releasing phenolic compounds located in the cell vacuoles [32]. Such effects have also been reported previously for other fruit juices [45,46,47]. It was obvious that evaporation methods significantly affected the antioxidant properties of date syrup samples. The improved antioxidant activities have also been reported in the coconut sap and sugarcane sugar produced using high processing temperatures [24,48,49]. This increment might be due to the formation of Maillard reaction products and caramelization products during the heating of coconut sap or sugarcane juice. Maillard reaction products are known to possess free radical scavenging activity, inhibiting lipid peroxidation [48]. TPC values of our date syrups were much lower but the FRAP values were higher than those reported previously [16,18], probably due to the differences in the varieties and maturity stages of dates. The very low DPPH values of date syrups could be explained by the fact that DPPH radical scavenging measured both the activities on hydrogen atom transfer and single electron transfer mechanisms, which were in the opposite trend [50]. Thus, it was possible that the DPPH assay is less sensitive than the FRAP and ORAC assays, which are based on either only hydrogen atom transfer or a single electron transfer reaction. Besides, it was previously suggested that the DPPH radical scavenging assay is more suitable for measuring antioxidant activity in the hydrophobic system [51].

Considering the physico-chemical and antioxidant properties of the syrup, open heating was considered the suitable evaporation condition for date syrup production. The properties of the obtained syrup did not much depend on the evaporation condition, except for color and antioxidant properties. Although vacuum evaporation could preserve the date syrup’s most remarkable quality characteristics, especially color, the syrup possessed inferior antioxidant properties to open heating. Moreover, evaporation using open heating would be more economical and more feasible for scaling up the production, particularly in the micro- and small food processing enterprises. Therefore, the syrups prepared using juices extracted without the aid of enzymes, and with the aid of Pectinex and Viscoszyme, followed by open heating evaporation, namely CO, PO, and VO, were selected for further analysis.

### 3.4. Chemical Composition

The nutritional composition of date syrups is listed in Table 4. CO had the highest moisture content, followed by VO and PO, respectively. The protein content of CO and PO were approximately similar and slightly lower than that of VO. Fat has been detected in deficient amounts in all syrups. Carbohydrate was the predominant component in all date syrups, accounting for 61–69% *w/w*. PO had the highest carbohydrate content, followed, respectively, by VO and CO. All syrups contained less than 1% w/w dietary fiber, most of which was soluble dietary fiber. Particularly, insoluble fiber was absent in VO. For all date syrups, sugars are the major carbohydrate contributor. PO had the highest sugar content, owing to its highest carbohydrate among the three date syrups. Considering the sugar composition, the sugar of all date syrups was composed mainly of glucose and fructose. Sucrose was only found in CO and none of the syrups contained maltose. The caloric value of syrups was corresponding to their carbohydrate, dietary fiber, and sugar contents. CO provided the least energy content of about 250 kcal/100 g, while PO and VO would give higher calories. However, the proximate composition of date syrups was within the range reported in the literature [7]. The slightly lower amount of soluble and insoluble dietary fibers in PO and VO could be the result of enzymatic hydrolysis of indigestible polysaccharide molecules into smaller molecules that are more soluble in water, or even more digestible. This also increased the amount of total sugar and reducing sugar content of PO and VO than in CO.

The presence of protein and reducing sugars (glucose and fructose) in all syrup samples assured that the Maillard reaction could occur during syrup production. Nevertheless, the more reducing sugars in PO and VO did not correspond to a reduction in the extent of the Maillard reaction, as evident in their lower BI (Table 1). On the other hand, sucrose’s presence in CO could be attributed to sugar crystallization, resulting in the higher viscosity and more shear thinning characteristics of CO than other syrups (Table 2). It was found that the glucose to fructose ratio of CO, PO, and VO were 1:0.92, 1:0.93, and 1:0.96, respectively. A glucose to fructose ratio close to 1:1 has been proven to be associated with the low glycemic index of sugars [52]. All date syrup samples contained HMF at 23–25 mg/kg (Table 4). The highest HMF was found in VO, while those of CO and PO were similar. The content was much below the maximum level allowed in honey and syrup by the Codex Standard, which is 40 mg/kg. A previous study has demonstrated that the HMF of palmyra palm sugar and syrup increased with the processing temperature [53]. It has also been reported that HMF in honey and palm syrup also provided antioxidant activities, though its health benefits were unclear [54]. Thus, HMF could possibly have an impact on the greater antioxidant activities of syrups evaporated by open heating than those produced by vacuum evaporation (Table 3).

### 3.5. Carbohydrate Digestive Enzyme Inhibitory Activities

The key enzymes related to diabetes, namely α-amylase and α-glucosidase, are important in the hydrolysis of polysaccharides into monosaccharides, which can be absorbed into cells. Inhibition of those enzymes might retard the digestion of carbohydrates and consequently delay the increase in the blood glucose level. In this study, the inhibitory activities toward the two enzymes were assessed using 50 μL date syrup in a 200 μL assay. PO provided a four-fold lower α-amylase inhibitory activity than VO and CO (Table 5). Between the two date syrups prepared using different enzyme-assisted extractions, PO possessed slightly greater activity for α-amylase inhibition than VO. All syrups exhibited 75% inhibition of α-glucosidase activity, and the differences among samples were very little. Enzyme inhibition could occur via different modes, including competitive inhibition and non-competitive inhibition. Competitive inhibitors compete with substrate in binding at the active site of enzyme, while non-competitive inhibitors do not bind to a site other than the active site and change its conformation [55]. Phytochemicals naturally present in date fruits, as well as the substance formed upon the production of date syrup, should be responsible for the inhibition of α-amylase and α-glucosidase. The inhibition on carbohydrate digestive enzymes of date seeds have also been found to be associated with their antioxidant activities and TPC [21,56].

### 3.6. Thermal Properties

Thermal properties are widely used to evaluate thermal stability of syrup and honey. Glass transition and melting behavior of date syrups were determined from DSC thermograms (Figure 4). For all syrups, three thermal phenomena were observed, including: (i) glass transition between −60 and −55 °C; (ii) a weak endothermic phenomenon at 45–60 °C caused by the melting of the water/sugar/starch complex or by sugar polymorphism; and (iii) a wide intense endothermic peak at 100–140 °C corresponding to the melting of sugars [57,58].

The onset of glass transition for all date syrups took place at −57 °C and reached its peak between −54 and −55 °C (Table 6). Such values were comparable to those reported previously for date syrups and honey [29]. Among the three syrups, no significant difference was not found in *T_g_* and *T_m_*. The *T_g_* value of syrups depends largely on the composition of sugar and the water content [29,57]. The increase in water content led to the lower *T_g_* value of syrup. It is plausible that the *T_g_* value of CO was the lowest among other samples because it had the highest moisture content, although the difference was not significant. The *T_m_* of all date syrup samples were between 127 and 134 °C. The melting peak of CO was broader and less intense than PO and VO. It has been reported that the melting curve of honey depended on its chemical composition and thermal history [57]. The unique and sharp melting point of date syrups prepared with enzyme-assisted juice extraction was similar to those of honey and sugar beet syrup [57], as well as the mixture of fructose, glucose, and sucrose [58].

### 3.7. Sensory Acceptability

Sensory attributes of date syrups, including appearance, color, odor, consistency, and overall acceptability, were evaluated using a 9-point hedonic scale. The panelists were aged 24–58 years old with the average age of 36.60 ± 11.22 years. It has been reported that people in the 20–59 years age group account for more than half of the Thai population, and they are major consumer group of syrups and spreads in Thailand [59].

The sensory acceptability scores for all attributes of date syrups ranged between 6 and 7, which was like slightly to like moderately (Figure 5). There was no significant difference among the sensory scores of the syrup samples. This means the three date syrups was equally accepted in terms of sensorial quality. Although their color and rheological properties were obviously different (Table 1 and Table 2), it did not affect the preference of the panelists. It could be that panelists could not detect the difference between the samples, or the difference was detectable but still acceptable. It is noteworthy that the standard deviations of all sensory acceptability scores were somewhat broad, with the coefficient of variation ranged from 20.74 to 28.85%. The wide age range of the panelists could be one of the main factors contributing to such variations. In particular, preference for sweet taste declines with age [60].

Taking all the results into account, PO had the greater antioxidant properties and the stronger inhibition on carbohydrate digestive enzyme than the syrup prepared without the aid of enzymes. Although most of the properties of PO were comparable with VO, the Pectinex-assisted juice extraction took a shorter time to reach its maximum soluble solids yield than Viscozyme. Therefore, only the date syrup PO, which was prepared using Pectinex-assisted extraction of juice followed by evaporation using open heating, was selected for further investigation.

### 3.8. Glycemic Properties

Due to the unusually low blood glucose level obtained in the oral glucose tolerance test, one out of 12 participating subjects was excluded. Finally, 11 healthy subjects (four males and seven females) aged 23–34 years with normal BMI were included in further analyses (Appendix A). In all subjects, the blood glucose level initially increased after 15 min of ingesting the tested foods, reached the highest peak at 30 min, and gradually declined to baseline at 120 min (Figure 6a). The postprandial glucose level of the date syrup at any durations after consumption was lower than the reference glucose. The IAUC of date syrup PO was significantly lower than glucose (Figure 6b).

The GI value was calculated from the IAUC of a tested food’s blood glucose response curve and reference food. GI describes the rate that carbohydrate is converted into glucose and absorbed. It was found that the date syrup PO was classified as low GI (55.13 ± 17.55), according to American Diabetes Association [61]. When GL was calculated from the amount of available carbohydrate in a standard serving of syrup at 30 g and its GI, the date syrup PO had the GL of 11.29 ± 3.59. Such GL value fell into the medium GL range, i.e., 11–19, owing to the high carbohydrate content of the date syrup (Table 4). The large variations in postprandial blood glucose level (2.97–33.74% coefficient of variation) were caused by variation among the human subjects, although their body mass index and fasting blood glucose level did not vary much (Appendix A). Similar findings are also evident in other in vivo studies of GI in foods [62,63].

The GI values of date syrups in this study were slightly higher than that previously reported for Barhi date fruits from the United Arab Emirates (GI = 49.7) [64]. It could be that the natural form of date fruits contained higher complex carbohydrates and dietary fibers than date juice and syrup, of which the fiber residue was removed. The fact that glucose to fructose ratio of PO was 1:0.93, which was closed to 1:1 (Table 4), could be another plausible reason. GI is known to be most influenced by the glucose to fructose ratio. Food containing more glucose tends to raise the plasma glucose level more rapidly than fructose, leading to a higher GI. Previous studies also found that maple syrup, honey, and agave syrup, of which the glucose to fructose ratios were 1:1, 1:1.27, and 1:1.05, respectively, exhibited low GI [52]. It was plausible that enzyme-assisted juice extraction enhanced the release of substances from the degraded tissues, which might increase the amount of indigestible or slow-digestible fractions or substances that could retard the digestion in the obtained syrups. This is consistent with the inhibitory activities on carbohydrate-digestive enzymes of the date syrup (Table 5). Moreover, phenolic content was found to be a significant factor governing the in vivo GI of carbohydrate-rich food [65], which corresponded well to the TPC of PO (Table 3). Hence, the digestion and the glucose release were diminished, resulting in the low GI of date syrup.

## 4. Conclusions

This study investigated various properties of date syrups prepared using different methods for juice extraction and evaporation. It was evident that the juice extraction method impacted the production yield, rheological properties, antioxidant properties, carbohydrate composition, and inhibitory activities on carbohydrate-digestive enzymes of the obtained syrups, while the evaporation method had a strong influence on the syrup color. However, such differences did not affect the HMF content, thermal properties, and consumer sensory acceptability of the syrups prepared with different extraction methods followed by evaporation using open heating. The results from the in vivo study indicated that the date syrup prepared from juice extracted with the aid of Pectinex followed by evaporation using open heating were low GI and medium GL. Thus, enzyme-assisted extraction of date juice using Pectinex could be an option to produce a healthy natural sweetener from date fruits, which would add the value of low-quality date fruits and help to reduce food loss in the dates supply chain. Further analyses should be conducted to elucidate the bioactive compounds and physiological effects of the syrups.

## Figures and Tables

**Figure 1 foods-12-01268-f001:**
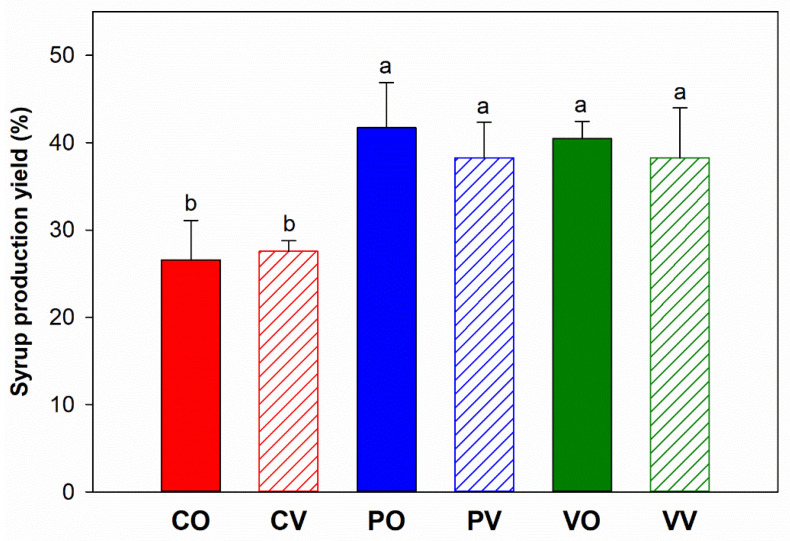
Production yield of date syrups prepared from juice extracted without enzyme (red-color bars: CO, CV), and with the aid of Pectinex (blue-color bars: PO, PV) or Viscozyme (green-color bars: VO, VV) followed by evaporation using open heating (filled bars: CO, PO, VO) or vacuum evaporation (diagonal striped bars: CV, PV, VV). Values are means and standard deviations of triplicate sample (*n* = 3). Different letters indicate significant difference between means (*p* < 0.05).

**Figure 2 foods-12-01268-f002:**
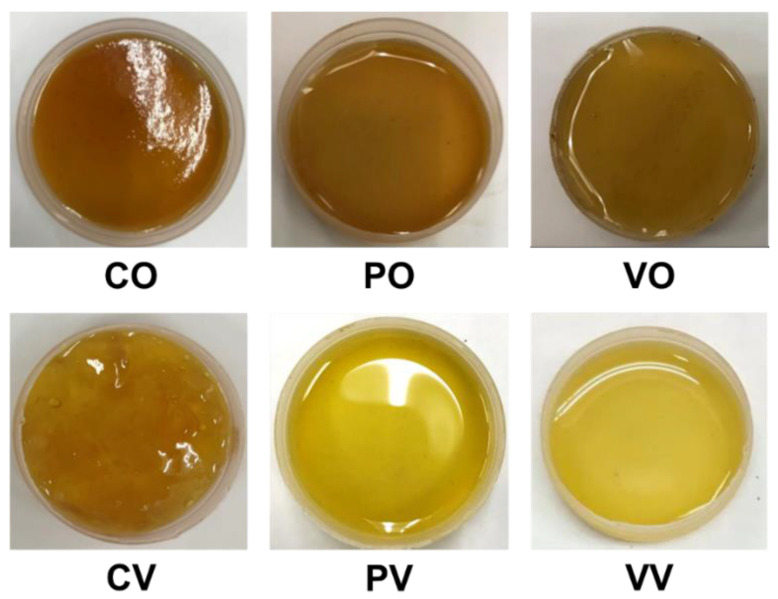
Photographs taken of date syrups prepared from juice extracted without enzyme (CO, CV) and with the aid of Pectinex (PO, PV) or Viscozyme (VO, VV) followed by evaporation using open heating (CO, PO, VO) or vacuum evaporation (CV, PV, VV).

**Figure 3 foods-12-01268-f003:**
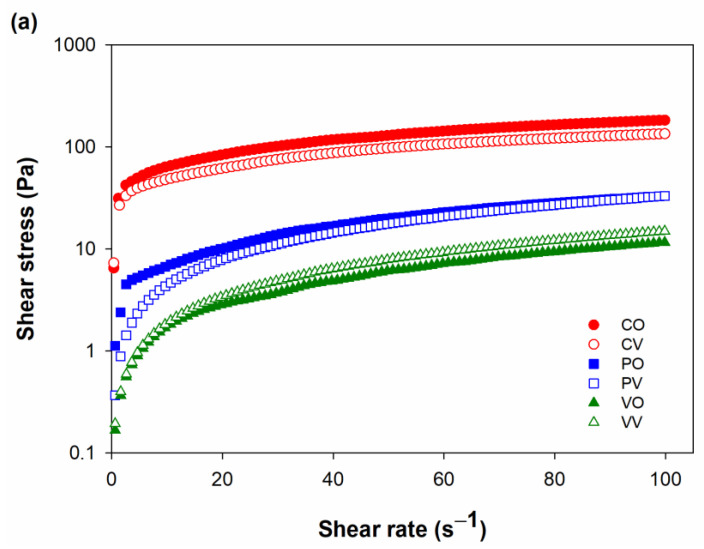
Relationship between shear rate and (**a**) shear stress and (**b**) viscosity measured at 25 °C of date syrups prepared from juice extracted without enzyme (●, ○) and with the aid of Pectinex (■, ☐) or Viscozyme (▲, △) followed by evaporation using open heating (●, ■, ▲) or vacuum evaporation (○, ☐, △).

**Figure 4 foods-12-01268-f004:**
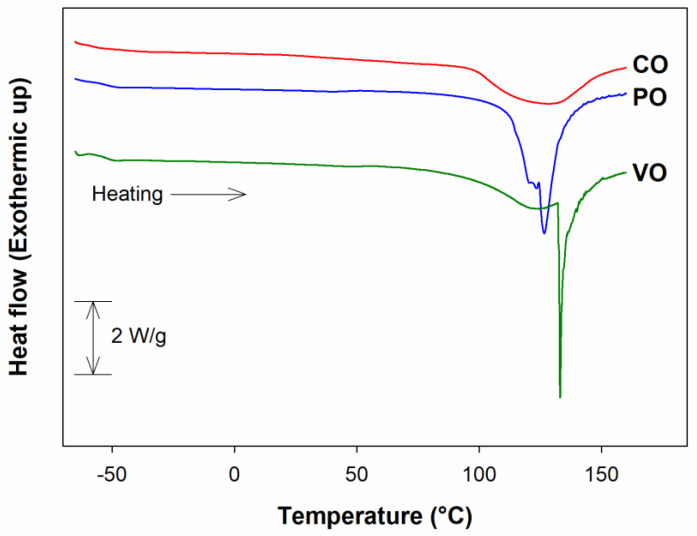
Typical DSC thermograms of date syrups prepared from juice extracted without enzyme (CO) and with the aid of Pectinex (PO) or Viscozyme (VO) followed by evaporation using open heating.

**Figure 5 foods-12-01268-f005:**
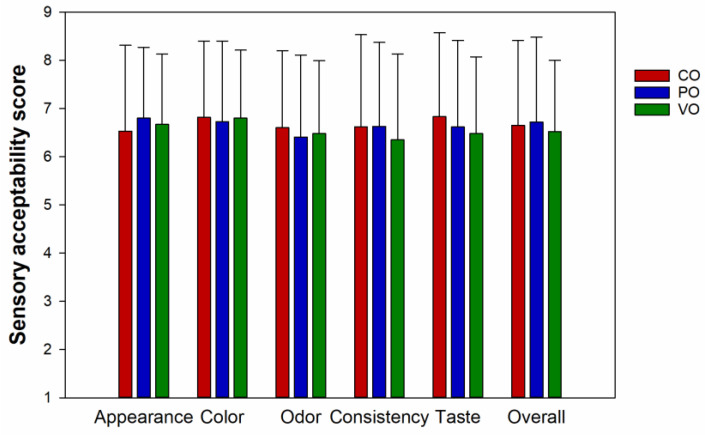
Sensory acceptability score of date syrups from juice extracted without enzyme (CO) and with the aid of Pectinex (PO) or Viscozyme (VO) followed by evaporation using open heating. Values are means and standard deviations of 60 panelists (*n* = 60). There was no significant difference between means for each sensory attribute of the 3 samples (*p* > 0.05).

**Figure 6 foods-12-01268-f006:**
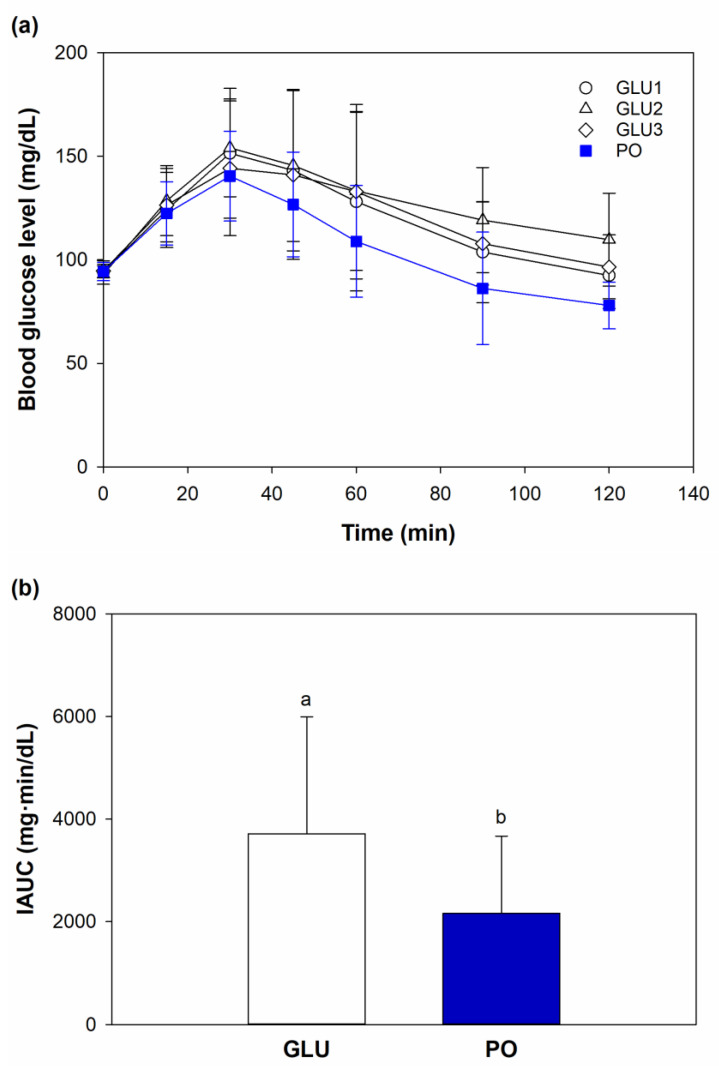
(**a**) Changes in blood glucose level and after ingestion of glucose standard as a reference food (GLU1, GLU2, GLU3), and date syrup prepared from juice extracted with the aid of Pectinex followed by evaporation using open heating (PO); (**b**) Incremental area under the curve after in-gestion of glucose standard (GLU) and date syrup (PO). Values are means and standard deviations of 11 subjects (*n* = 11). Different superscripts indicate significant difference between means (*p* < 0.05).

**Table 1 foods-12-01268-t001:** Physicochemical properties of date syrups prepared using different extraction and evaporation methods.

Samples	Color Values ^†^			BI ^†^	ΔE* ^†^
	L*	a*	b*		
CO	13.57 ± 0.23^c^	16.36 ± 0.25 ^a^	20.64 ± 0.14 ^d^	584.73 ± 30.45 ^a^	0.00 ^d^
CV	27.85 ± 0.18 ^a^	1.57 ± 0.27 ^d^	33.96 ± 0.10 ^a^	341.52 ± 6.70 ^c^	24.46 ± 0.08 ^a^
PO	14.58 ±0.33 ^b^	13.27 ± 0.26 ^b^	19.30 ± 0.48 ^e^	428.11 ± 29.80 ^b^	3.65 ± 0.09 ^d^
PV	28.23 ± 0.85 ^a^	0.51 ± 0.05 ^e^	29.53 ± 0.44 ^c^	232.67 ± 20.16 ^d^	23.32 ± 0.49 ^b^
VO	14.50 ± 0.43 ^b^	11.38 ± 0.19 ^c^	19.36 ± 0.07 ^e^	438.36 ± 31.71 ^b^	5.37 ± 0.18 ^c^
VV	27.90 ± 0.61 ^a^	0.38 ± 0.05 ^e^	30.70 ± 0.05 ^b^	260.46 ± 12.65 ^d^	23.68 ± 0.36 ^b^

CO, CV, PO, PV, VO, VV: date syrups prepared from juice extracted without enzyme (CO, CV), and with the aid of Pectinex (PO, PV) or Viscozyme (VO, VV) followed by evaporation using open heating (CO, PO, VO) or vacuum evaporation (CV, PV, VV); ^†^ Means ± standard deviations of triplicate samples (*n* = 3); ^a–d^ Mean values within a column with different superscript letters are significantly different (*p* < 0.05).

**Table 2 foods-12-01268-t002:** Rheological properties of date syrups prepared using different extraction and evaporation methods.

Samples	Rheological Properties ^†^		
	Apparent Viscosity at 50 s^−1^ (mPa·s)	Flow Behavior Index	Consistency Index (Pa·s^n^)
CO	2472.73 ± 206.63 ^a^	0.45 ± 0.02 ^c^	21.53 ± 0.23 ^a^
CV	1997.68 ± 155.05 ^b^	0.44 ± 0.00 ^c^	18.42 ± 1.54 ^b^
PO	387.40 ± 10.50 ^c^	0.63 ± 0.00 ^b^	1.65 ± 0.02 ^c^
PV	331.92 ± 19.97 ^cd^	0.86 ± 0.01 ^a^	0.58 ± 0.03 ^cd^
VO	142.63 ± 33.30 ^d^	0.88 ± 0.02 ^a^	0.23 ± 0.04 ^d^
VV	153.28 ± 16.13 ^d^	0.86 ±0.03 ^a^	0.26 ± 0.00 ^d^

CO, CV, PO, PV, VO, VV: date syrups prepared from juice extracted without enzyme (CO, CV), and with the aid of Pectinex (PO, PV) or Viscozyme (VO, VV) followed by evaporation using open heating (CO, PO, VO) or vacuum evaporation (CV, PV, VV); ^†^ Means ± standard deviations of triplicate samples (*n* = 3); ^a–d^ Mean values within a column with different superscript letters are significantly different (*p* < 0.05).

**Table 3 foods-12-01268-t003:** Antioxidant properties of date syrups prepared using different extraction and evaporation methods.

Samples	TPC (mg GAE/ 100 mL) ^†^	Antioxidant Activities (µmol TE/100 mL) ^†^		
		DPPH (×10^−2^)	FRAP	ORAC
CO	5.36 ± 0.37 ^b^	5.68 ± 0.49 ^a^	31.36 ± 1.14 ^b^	79.00 ± 7.94 ^c^
CV	3.81 ± 0.30 ^d^	6.05 ± 0.49 ^a^	22.15 ± 2.19 ^d^	62.78 ± 5.22 ^d^
PO	7.13 ± 0.22 ^a^	2.06 ± 0.12 ^b^	38.84 ± 2.05 ^a^	109.93 ± 3.31 ^a^
PV	4.80 ± 0.25 ^c^	0.85 ± 0.09 ^d^	26.05 ± 2.52 ^c^	79.90 ± 2.60 ^c^
VO	6.69 ± 0.37 ^a^	1.51 ± 0.11 ^c^	31.97 ± 1.81 ^b^	96.56 ± 7.94 ^b^
VV	5.08 ± 0.27 ^bc^	0.77 ± 0.08 ^d^	27.72 ± 1.23 ^c^	63.31 ± 5.79 ^d^

CO, CV, PO, PV, VO, VV: date syrups prepared from juice extracted without enzyme (CO, CV), and with the aid of Pectinex (PO, PV) or Viscozyme (VO, VV) followed by evaporation using open heating (CO, PO, VO) or vacuum evaporation (CV, PV, VV); ^†^ Means ± standard deviations of triplicate samples (*n* = 3); ^a–d^ Mean values within a column with different superscript letters are significantly different (*p* < 0.05).

**Table 4 foods-12-01268-t004:** Chemical composition of date syrups prepared using different extraction methods followed by evaporation using open heating.

Chemical Composition	Amount ^†^		
	CO	PO	VO
Moisture (g/100 g)	36.16	27.63	29.70
Protein (N × 6.25) (g/100 g)	0.81	0.88	0.96
Total fat (g/100 g)	0.00	0.00	0.00
Ash (g/100 g)	2.97	2.36	2.05
Total carbohydrates (g/100 g)	61.49	69.14	67.31
Total dietary fibers (g/100 g)	0.98	0.90	0.87
Soluble dietary fiber (g/100 g)	0.90	0.89	0.87
Insoluble dietary fiber (g/100 g)	0.08	0.01	0.00
Total sugar (g/100 g)	58.35	71.27	65.52
Glucose (g/100 g)	30.39	36.95	33.39
Fructose (g/100 g)	27.96	34.32	32.14
Sucrose (g/100 g)	2.15	ND	ND
Maltose (g/100 g)	ND	ND	ND
Energy (kcal/100 g)	249.14	280.06	272.98
HMF (mg/kg)	23.55	23.01	25.75

CO, PO, VO: date syrups prepared from juice extracted without enzyme (CO) and with the aid of Pectinex (PO) or Viscozyme (VO) followed by evaporation using open heating; ND: Not detected; ^†^ Means of duplicate analyses of pooled samples.

**Table 5 foods-12-01268-t005:** Inhibitory activities on carbohydrate digestive enzymes of date syrups prepared using different extraction methods followed by evaporation using open heating.

Samples	Inhibitory Activities (% Inhibition) ^†,‡^	
	α-Amylase	α-Glucosidase
CO	5.91 ± 0.33 ^b^	74.25 ± 0.32 ^b^
PO	22.31 ± 1.80 ^a^	75.70 ± 0.38 ^a^
VO	19.83 ± 2.06 ^a^	74.59 ± 0.47 ^ab^

CO, PO, VO: date syrups prepared from juice extracted without enzyme (CO) and with the aid of Pectinex (PO) or Viscozyme (VO) followed by evaporation using open heating; ^†^ % inhibitory activity of 50 μL date syrup in a 200 μL assay; ^‡^ Means ± standard deviations of triplicate samples (*n* = 3); ^a, b^ Mean values within a column with different superscript letters are significantly different (*p* < 0.05).

**Table 6 foods-12-01268-t006:** Thermal properties of date syrups prepared using different extraction methods followed by evaporation using open heating.

Samples	Thermal Behavior ^†^	
	Glass Transition, *T_g_* (°C) ^NS^	Melting Point, *T_m_* (°C) ^NS^
CO	−55.37 ± 1.65	131.30 ± 7.19
PO	−53.43 ± 2.37	127.03 ± 1.02
VO	−54.70 ± 1.11	133.90 ± 4.65

CO, PO, VO: date syrups prepared from juice extracted without enzyme (CO) and with the aid of Pectinex (PO) or Viscozyme (VO) followed by evaporation using open heating; ^†^ Means ± standard deviations of triplicate samples (*n* = 3); ^NS^ Mean values within a column are not significantly different (*p* > 0.05).

## Data Availability

Data is contained within this article and Appendix A.

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
