# Peer review of "Effects of Extraction and Evaporation Methods on Physico-Chemical, Functional, and Nutritional Properties of Syrups from Barhi Dates (Phoenix dactylifera L.)"

_foods, 2023, doi:10.3390/foods12061268_

Round 1

Reviewer 1 Report

The manuscript, titled Effects  of  Extraction  and  Evaporation  Methods  on  Physicochemical, Functional, and Nutritional Properties of Syrups from Barhi Dates (Phoenix dactylifera L.) , raises an important topic related to preventing food waste by obtaining a food product with a wide range of uses. All comments are included below.

1. Subsection 2.1. Preparation of date syrup should be changed, because it is not description of syrup preparation process. This part is about materials.

2. Line 120 – 123: Analysis were made on fresh or frozen dates?

3. Line 124 -127: Please verify the given enzyme activities, they are not in accordance with the manufacturer's declaration. An enzyme from another supplier was probably used.

4. Line 135: Due to incorrect data on enzyme activity, it is necessary to verify what doses of enzymes were used during the extraction. Please explain on what basis the dose of 50 U/100 g date flesh was selected?

5. Line 138: How was the extraction time for individual enzymes selected?

6. Throughout the manuscript, please remove the space before symbols Celsius and Brix degree.

7. I suggest to organise the subsection on methods, because in 2.3 Determination of physico-chemical properties there are rheological properties, but there is no chemical composition etc.

8. There is no information about gap between cone and plate.

9. On what basis was the shear rate increase time selected?

10. Line 181: I suggest mPa·s instead of cP

11. The symbol of shear rate is gamma with dot.

12. Please explain how the yield stress was determined, because the description is unclear. I am asking for the characteristics of the conducted experiment with the use of a rheometer. The description indicate that experiment was performed in CR (controlled rate) conditions not in CS (controlled stress), which I designed to determine values of yield stress.

13.  Tab Table 2: The value of the yield point indicated for the CV syrup is not in accordance with the data presented in Figure 2(a). Please explain.

14. Line 338: Liquid is Newtonian or not. Please correct the sentence.

15.  Line 429 – 433: It should be noted that viscosity values are obtained at shear rate 50 s-1.

16. Line 441: The fluid cannot be less or more shear thinning. Please correct this sentence.

17. Line 440 – 446: In these lines Authors wrote about polysaccharides, absent in enzymatic extraction and present in conventional extraction. What polysaccharides may be present in the traditional extraction method and how can the equal amount of fiber in the various syrup variants be explained?

18. Line 444 – 446: The mere presence of the network does not cause shear thinning behaviour, but its deformation due to the applied shear rate. Please correct the sentence.

18. Only one publication from 41-43 describes shear thinning behaviour of date syrups. It should be verify.

19. Line 448: same remark as for Line 441

19. Please correct the units in Table 4.

Reviewer 2 Report

This article "Effects of Extraction and Evaporation Methods on Physicochemical, Functional, and Nutritional Properties of Syrups from Barhi Dates (Phoenix dactylifera L.)” was major revised and has not a novelty. I suggest that the following comments be considered.

Title: It is good.

Abstract:

·         The type of statistical design used in this research should be mentioned.

Keywords: Please choose keywords other than the main words of the title. In this case, other researchers can find your article by searching a wide range of words through databases. I propose other keywords as the follow: Pectinase, cellulose instead of enzyme-assisted extraction

Introduction:

·         Line 80 No comparison has been made in terms of extraction efficiency with other methods.

·         The enzyme extraction method is a pre-treatment, but you used it as a main extraction method

·         in keywords Enzymes-Assisted Extraction mentioned but the enzymatic method alone has been used

Materials:

·         In Line 127 Written All chemicals and reagents, unless stated otherwise, were obtained from Sigma-Aldrich (St. Louis, MO, U.S.A.) Please write materials as Company Name (City, Country), especially for chemical analysis assessment which used in the study.

Methodology:                         

·         enzymes concentration have not been specified

·         Pectinase and cellulase enzyme activities have not been reported

 Please use newer and appropriate references. As a suggestion: https://doi.org/10.1186/s40538-021-00220-z

 “Results:

·         Table 4. 6.: The alphabetical statistical letters for the means should all be modified such that the greatest number has the letter a and as the numbers go lower, letters b, c etc.

 Most of the tables have no statistical comparison for them

Discussion:

Discussion must improve and in some cases it is very weak

Conclusions:

Conclusion is very boring, try to make it more scientific, comprehensive and concise in detail, especially.

Reviewer 3 Report

Keywords: replace "date syrup" with "syrup".

 Line 71. Replace "... dates were used..." by "date have been used.."

The objective of the Abstract does not completely match the objective shown in line 100. The following change is suggested:

"This study aimed to investigate the effects of juice extraction and evaporation methods on the physico-chemical, functional, nutritional and sensory properties on the syrup prepared from Barhi dates"

 In section 2. Materials and Methods, add:

2.1. Materials and reagents

2.2. Preparation od date syrup

2.3. Preparation of date syrup

Line 112: What are the "company's quality specifications"?

Line 120. Add the Year of AOAC official methods.

Section 2.1. and 2.2. The title is repeated.

In section 2.2.1. On the basis that the substrate enzyme ratio was 50U/100g? Additionally, this section (line 131 to 144) absences references to support its methodology.

Section 2.2.2.  What was the process, open hearting, or vacuum evaporation?  What is the advantage of each, and the disadvantage?  Explain in writing. If both were used, for what? There is a absence of references to support the methodology.

Section 2.3.1. reference. Add how many replicas were made.

Section 2.3.2. reference. Add if replicas were made.

Line 190. Delete "our".

Line 212. Add the conversion factor used in the Protein content.

Section 2.7. Add Reference

Based on which standard or article, was the sensory evaluation performed? Why is bias from 22 to 60 years? I think it is very broad, or perform several groups, but more biased, to avoid variability.

Section 2.9. References

Section 3.1. Explain in detail the effect that enzymes have on the microstructure-composition of tissue.

Given the importance of color, and that it is related to the processes to which syrups are subjected, I suggest that Figure S1 be placed within the article. It is recommended that the DE determined with the CIELab.

In Table 1, they mean CO, CV, PO, PV, VO, VV, etc... Place at the bottom of the figure. Likewise, within the writing.

Line 381: relate the BI of the syrups with the pH.

I suggest a rearrangement in the themes, first it would be convenient to talk about the chemical composition, and then in the other properties.

Figure 4, it is important to review that they present a wide standard deviation, this may be due to several factors, and one of the main ones could be associated with the wide age range.

In Figure 5a, why is the standard deviation so wide?

Round 2

Reviewer 1 Report

Please correct these two sentences. A statement about the degree of shear thinning cannot be used because the viscosities and stresses are different for the tested syrups. Please use the value of coefficient n.

462-463 " and exhibited a lesser degree of shear thinning behavior (higher flow behavior index of closer to 1) than the juice extracted without enzyme"

471-473 "The lower viscosity and lesser degree of shear thinning behavior of date syrups prepared by enzyme-assisted extraction of date juice might be advantageous for their application in food"

Reviewer 2 Report

The manuscript has been revised according to comments.